# The Distribution Characteristics of a 19-bp Indel of the *PLAG1* Gene in Chinese Cattle

**DOI:** 10.3390/ani9121082

**Published:** 2019-12-04

**Authors:** Zihui Zhou, Bizhi Huang, Zhenyu Lai, Shipeng Li, Fei Wu, Kaixing Qu, Yutang Jia, Jiawen Hou, Jianyong Liu, Chuzhao Lei, Ruihua Dang

**Affiliations:** 1Key Laboratory of Animal Genetics, Breeding and Reproduction of Shaanxi Province, College of Animal Science and Technology, Northwest A&F University, Xianyang 712100, China; 15636110299@163.com (Z.Z.); lzy18408210600@126.com (Z.L.); lsp17782767206@163.com (S.L.); 18392360732@163.com (F.W.); houjiawen@163.com (J.H.); leichuzhao1118@126.com (C.L.); 2Yunnan Academy of Grassland and Animal Science, Kunming 650212, China; hbz@ynbp.cn (B.H.); Kaixqu@163.com (K.Q.); ljy@ynbp.cn (J.L.); 3Institute of Animal Science and Veterinary Medicine, Anhui Academy of Agriculture Science, Hefei 230001, China; yutang2018@163.com

**Keywords:** cattle, growth traits, indel, *PLAG1* gene, polymorphism

## Abstract

**Simple Summary:**

Growth traits are important quantitative traits for cattle performance, which influences herd productivity. Previous studies demonstrated that a 19-bp indel in pleomorphic adenoma gene 1 (*PLAG1*) is associated with bovine growth traits. In our studies, the distribution of the 19-bp indel of *PLAG1* gene among 37 Chinese cattle breeds and the association between this indel and growth traits of Yunling cattle were explored. As a result, we found that the average body height of various Chinese cattle breeds showed an increasing trend from south to north of China, which was consistent with the distribution trend of the 19-bp indel in cattle at different geographic latitudes. Moreover, this indel was verified to be significantly associated with the growth traits of Yunling cattle. Our results showed that the 19-bp indel in *PLAG1* strongly influences body size in Chinese cattle.

**Abstract:**

Pleomorphic adenoma gene 1 (*PLAG1*) belongs to the *PLAG* family of zinc finger transcription factors. In cattle, a 19-bp insertion/deletion (indel) was identified in intron 1 of the *PLAG1* gene (GenBank Accession No. AC_000171.1). Researches showed that the indel is polymorphic in Chinese cattle breeds such as Qinchuan cattle, Pinan cattle, Xianan cattle, and Jiaxian red cattle, and correlation analysis showed that the polymorphism is related to the height of these cattle breeds. Chinese cattle breeds show a difference in height related to geographical distribution. We investigated the distribution of the 19-bp indel polymorphism in 37 cattle breeds, including 1354 individuals. The results showed that there were three genotypes and two alleles (W, 366 bp; D, 347 bp). From northern cattle to southern cattle, the frequency of W allele gradually decreased, while the frequency of D allele showed an opposite trend, which was consistent with the distribution of cattle breeds of different height in China. Therefore, the polymorphism of this indel may be related to the regional distribution of cattle breeds in China. In addition, we chose Yunling cattle with a mixed genetic background to study the genetic effects of the 19-bp indel on body size traits. Statistical analysis showed that *PLAG1* was significantly associated with the body height, cross height, and chest circumference of Yunling cattle (*p* < 0.05). This study provides new evidence that the 19-bp indel of the *PLAG1* gene is a highly effective trait marker that can be used as a candidate molecular marker for cattle breeding.

## 1. Introduction

As a kind of livestock with important economic value, cattle play an indispensable role in animal husbandry. According to geographical distribution, cattle in China are divided into central cattle (Qinchuan cattle, Nanyang cattle, Luxi cattle, Jinnan cattle, etc.), northern cattle (Mongolian cattle, Yanbian cattle, and Kazakh cattle), and southern cattle (Ji’an cattle, Jinjiang cattle, Wannan cattle, Weining cattle, etc.) [1]. Due to regional differences, there are significant differences in the body height traits of cattle breeds in different regions. The application of molecular marker polymorphisms and other techniques can provide a basis for the classification of Chinese cattle and the screening of body height traits [2].

Pleomorphic adenoma gene 1 (*PLAG1)* is involved in cell proliferation by directly regulating a wide array of target genes, including a number of growth factors such as *IGF2* [3]. The *PLAG* family is composed of *PLAG1, PLAGL1/LOT1/ZAC1,* and *PLAGL2*, and its structure and function are highly conservative [4]. Genetic studies found that *PLAG1* regulates cell apoptosis and gap 1 (G1) cell-cycle arrest, and it is associated with the development of a variety of tumors such as adioblastoma. Genome-wide association analysis showed that single nucleotide polymorphisms (SNPs) located in the *PLAG1* gene region were closely related to adult height [5]. Also, *PLAG1* is an important candidate gene affecting the growth and development of domestic animals. Some studies showed that the *PLAG1* gene plays a certain regulatory role in milk production, reproductive performance, muscle formation, and body height of livestock [6,7]. In pigs, the *PLAG1* gene has a certain effect on the length of limb bones [8]. In mice, knockout of the *PLAG1* gene showed that growth and development were blocked [9]. In addition, it was found that the 19-bp insertion/deletion (indel) polymorphism (rs523753416) detected on the intron of chromosome 14 of *PLAG1* gene was widespread in Qinchuan cattle [10], Pinnan cattle, Xia’ nan cattle, Jiaxian red cattle, and other Chinese cattle, and it was related to the height of cattle, but the frequency distribution was quite different.

Based on this background, determining whether the 19-bp indel affected the body height between northern and southern cattle breeds in China was the aim of our study. The polymorphism of the 19-bp indel in different areas of China was investigated, and its relationship with the growth and development of Yunling cattle was analyzed to verify its extensive effects. The detection of genes potentially associated with economic traits and the identification of effective mutations can provide a basis for the molecular marker-assisted selection of livestock.

## 2. Materials and Methods

### 2.1. Animals and Data Collection

The study sample had a total of 1354 individuals, 37 breeds of adult cattle, including three northern breeds (*N* = 81), seven Central Plains breeds (*N* = 180), 23 southern breeds (*N* = 555), two commercial breeds (*N* = 463), and two breeds for control (*N* = 48) and Tibetan cattle (N = 27). The distribution of the 37 cattle breeds of China, as well as Angus and Holstein populations, is shown in Appendix A.

According to the improved phenol–chloroform method [11], genomic DNA of 1354 individuals representing 37 breeds of Chinese cattle, Angus cattle, and Holstein cattle was extracted from bovine blood, and the growth data of 451 Yunling cattle individuals were quantified, including 10 body size indexes such as body height, hip cross height, body length, heart girth, chest width, rump length, chest depth, hip width, hucklebone width, and hip circumference, which were measured according to the method described by Gilbert et al. [12].

### 2.2. Primer Design and PCR Amplification

The insertion/deletion site (No. AC_000171.1 25020872-25020890) was genotyped using a pair of primers. The primers were designed on the National Center for Biotechnology Information (NCBI) Primer BLAST and synthesized by Shanghai Sangon Biotech Engineering Co., Ltd. The purification method was High Affinity Purification (HAP). A total of 2 OD divided into two tubes was provided. the size and sequence of the primers, the product fragment size, and Tm (°C) are shown in Table 1.

The PCR reaction system was 12.5 μL, and the contents of each component were as follows: 10 pmol of each primer, 6.25 μL of 2× PCR mix (Kangwei century biotechnology co., LTD, Beijing, China), 0.5 μL of each primer, 10 ng of genomic DNA, and 4.75 μL of ddH_2_O. The procedure of PCR amplification was as follows: pre-denaturation at 95 °C for 5 min, denaturation at 95 °C for 30 s, annealing at 57 °C for 30 s, extension at 72 °C for 30 s, 35 cycles, and extension at 72 °C for 10 min. The PCR instrument was from Hema Medical Instrument Co., Ltd (GuangDong, China). PCR amplification products were genotyped by 10% polyacrylamide gel electrophoresis at 220 V for about 2 h [13]. A total of 10 PCR products were selected and sent to Shanghai Sangon Biotech (Shanghai, China) for sequencing. The instrument for sequencing was ABI 3730xl in Shanghai Sangon Biotech (Shanghai, China).

The allele frequency and genotype frequency were determined according to the method of Maeiulla et al. The calculation formulas were as follows [14]:Genotype frequency (FA*i*A*j*) = A*i*A*j* number of individuals/total number of samples;(1)
Allele frequency (FA*i*) = FA*i*A*j* + 1/2 FA*i*A*j*.(2)
The Hardy–Weinberg equilibrium (HWE) [15] was tested, and population genetic indices such as gene heterozygosity He, gene homozygosity Ho (He + Ho = 1), effective allele number (Ne), and polymorphism information content (PIC) were calculated.
(3)$$H_{0}=\overset{n}{\underset{i=1}{\sum }}P_{i}^{2}$$
(4)$$H_{e}=1-\overset{n}{\underset{i=1}{\sum }}P_{i}^{2}$$
(5)$$N_{e}=1/\overset{n}{\underset{i=1}{\sum }}P_{i}^{2}$$
(6)$$PIC=1-\overset{m}{\underset{i=1}{\sum }}P_{i}^{2}-\overset{m-1}{\underset{i=1}{\sum }}\overset{m}{\underset{j=i+1}{\sum }}2P_{i}^{2}P_{j}^{2}$$
The association between the 19-bp indel locus in the *PLAG1* gene of Yunling cattle and its partial height traits was analyzed by SPSS 19.0 software (Statistical Product and Service Solutions, Version 19.0 Edition, IBM, Armonk, NY, USA) for one-way analysis of variance (ANOVA). The linear model used was as follows:Y*ij* = μ + A *i* + I *j* + E *ij*(7)
where Y*ij* is the trait measured on each of the *ij*-th animals, μ is the overall population mean of each trait, A*i* is fixed effect due to the *i*-th age, I*j* is the fixed effect associated with the *j*-th genotype, and E*ij* is the random residual error. The data were corrected by the least-squares method, and the genotype frequency and allele frequency were obtained according to the above method using Popgene software [16].

## 3. Results

### 3.1. PCR Amplification and Polyacrylamide Gel Electrophoresis

According to the primer of the *PLAG1* gene designed by Sangon Biotech, the length of the PCR amplified fragment was predicted to be 366 bp.

PCR amplification products were genotyped by 10% polyacrylamide gel electrophoresis. The bands of the electrophoresis were clear and bright, the length of the bands was consistent with the prediction, and three genotype bands were present, which were WW, DD, and WD. The sequencing results of this site are also consistent with the results displayed by the running gel (Figure 1).

### 3.2. Regional Difference of Allele Frequency

According to the regional information of the breeds classified in Animal Genetic Resources In China-Bovines, the results showed that, in the northern group, the frequencies of the W and D alleles were 0.78 and 0.22, respectively, and the frequency of allele W accounted for the majority. In the central group, the frequencies of the W and D alleles were 0.51 and 0.49, respectively, and the frequencies of the two alleles were nearly equal. In the southern group, the frequencies of the W and D alleles were 0.31 and 0.69, respectively, and the frequency of the W allele was lower. Similar to the results of the northern and central groups, in the commercial group, the frequencies of the W and D alleles were 0.59 and 0.41, respectively, and the frequency of the W allele was also higher than the D allele. The geographical distribution and allele frequencies of each breed are shown in Figure 2. In addition to the 35 domestic breeds shown in the figure below, for the allele frequencies of two foreign breeds used as the control, Angus cattle and Holstein cattle, the frequencies of the W allele were 0.68 and 0.65, respectively, and the frequencies of the D allele were 0.32 and 0.35, respectively. The results of genotypic and allele frequencies of the *PLAG1* gene across the 37 cattle breeds are shown in Appendix A. Meanwhile, the average body height of these cattle breeds is also given in Appendix A [17].

### 3.3. Verification of Population Genetic Effects of PLAG1 Gene in Yunling Cattle

After analysis of genetic polymorphism indicators such as genotype, genotype frequencies, allelic frequencies, heterozygosity (He), effective allele (Ne), and the polymorphic information content (PIC) of the *PLAG1* polymorphism, the results are shown in Table 2 and Table 3. According to the Botstein polymorphic information content classification standard, Yunling cattle had medium genetic diversity in the indel locus [18]. The point belongs to a moderate polymorphism, and the dominant genotype of the *PLAG1* gene is WD, whereas the dominant allele is W, and the polymorphic information content is 0.37. By using SPSS (19.0) software, a one-way analysis of variance (ANOVA) of the Yunling cattle *PLAG1* gene and its height traits was performed. The polymorphism of this gene was significantly correlated with body height, hip cross height, and chest circumference of the Yunling cattle population, as shown in Table 4.

## 4. Discussion

The 19-bp indel analyzed in the cattle *PLAG1* gene in this study is located in the intron region. Although introns are non-coding regions, they contain many cis-acting elements that bind to trans-acting factors to regulate gene transcription. Intronic mutations often act as enhancers or silencers to regulate gene transcription [19]. Also, intronic mutations can affect protein expression by affecting the shear and stability of target gene messenger RNA (mRNA), or by imbalanced chain intronic mutations and causal mutations [20]. Polymorphisms found in the intron region of the human *CXCR3* gene were verified to inhibit gene expression [21]. Similarly, the 20-bp indel polymorphism identified in the sheep *PRNP* gene also affected the expression of other growth-related genes [22].

The northern and central cattle are mainly found in the high latitudes of China and are relatively tall compared to the lower latitudes in the south. Our results showed that the distribution of allele frequency of the northern and central cattle was related to the distribution of their body height. With the increase in the latitude of cattle breeds, the W allele frequency also showed an increasing trend. This result was also reflected in the control groups. One of the controls, Holstein cattle, is native to the province of North Holland and West Friesland in the north of the Netherlands [23] (latitude information: 49°–53°). The height of adult Holstein cattle can reach 155–175 cm. Meanwhile, the height of the control Angus cattle (latitude information: 50–58°) was also higher than that of most cattle breeds in China [24]. In the results, we also found that the body height of the central group cattle is higher than the northern group (Appendix A), which may be related to hybrid vigor and resource environment in the Central Plains [17].

In order to verify its wide effect, the 19-bp indel in 27 Tibetan cattle was detected in this study, and it was found that the allele frequencies of W and D were 1 and 0, respectively. However, according to the body size data, Tibetan cattle are smaller than other breeds in size. This may be related to the high-altitude hypoxic environment of Tibetan cattle and their isolation from other Chinese cattle species.

In addition, in order to further verify the reliability of this marker, the variants of this polymorphism and its influence on growth traits were also detected in Yunling cattle, which has a mixed genetic background bred from brahmin cattle, Murray grey cattle, and Yunnan cattle by ternary hybridization.

The results showed that the polymorphism of the *PLAG1* gene 19-bp indel in Yunling cattle was significantly correlated with its body size, indicating that the gene locus had a strong population genetic effect. However, in order to explore the mechanism of the association between the indel and cattle growth traits, further studies are still needed on the internal mechanism of the *PLAG1* gene at the transcription and translation levels.

## 5. Conclusions

Our research investigated the distribution of the 19-bp indel polymorphism of the *PLAG1* gene in 37 cattle breeds, including 1354 individuals, and we found that, in this locus, the frequencies of the W and D alleles show a regular distribution from the northern cattle groups to the southern cattle groups, consistent with their different height in China. Furthermore, we found that, in Yunling cattle, the 19-bp indel polymorphism of *PLAG1* was significantly correlated with body height, cross height, and chest circumference (*p* < 0.05). These results suggested that the 19-bp indel of the *PLAG1* gene is an effective marker for growth traits in Chinese cattle.

## Figures and Tables

**Figure 1 animals-09-01082-f001:**
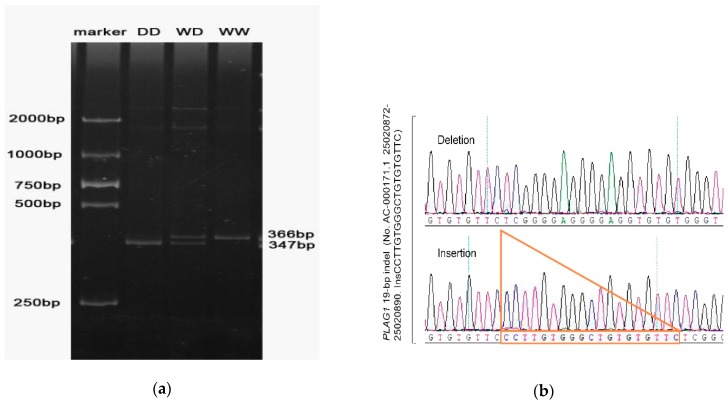
(**a**) The result of polyacrylamide gel electrophoresis. (**b**) The results of sanger sequencing.

**Figure 2 animals-09-01082-f002:**
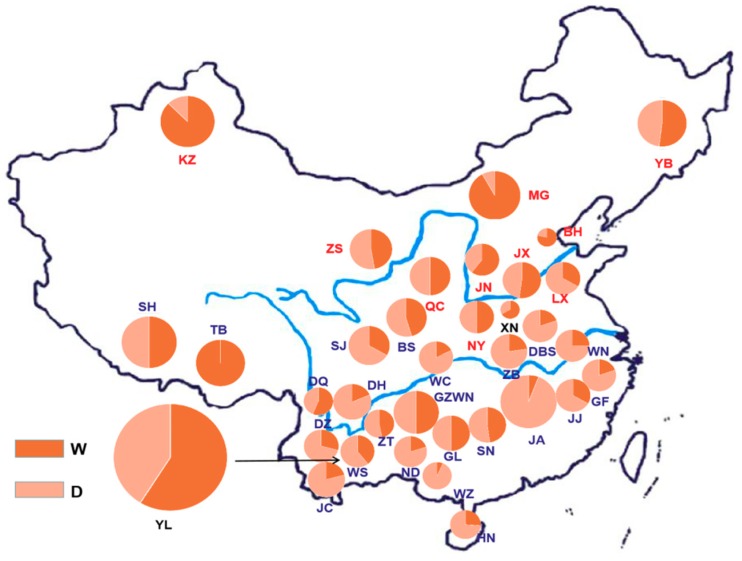
Distribution of allele frequency of the *PLAG1* gene in Chinese breeds. BH, Bohai Black; BS, Bashan; DBS, Dabeishan; DH, dehong; DQ, Diqing; DZ, Dianzhong; GF, Guangfeng; GL, Guanling; GZWN, Weining; JA, Ji’an; JC, Jiangcheng; JJ, Jinjiang; JN, Jinnan; JX, Jiaxian red; KZ, Kazakh; LX, Luxi; MG, Mongolian; ND, Nandan; NY, Nanyang; QC, Qinchuan; SJ, Sanjiang; SH, Shigatse Humped; SJ, Sanjiang; SN, sinan; TB, Tibetan; WC, Wuchuan; WL, Wuling; WN, Wannan; WS, Wenshan; XN, xianan; YB, Yanbian; YL, Yunling; ZB, Zaobei; ZS, Zaosheng; ZT, Zhaotong.

**Table 1 animals-09-01082-t001:** Information on primer sets used in this study. F—forward; R—reverse.

Primer Sequences (5′–3′)	Primer Size (bp)	Product Length (bp)	Tm (°C)
F: AAAAGAGTCCGCGTTTACTGC	21	366/347	57
R: CGATGAACTCTCCACCTGCG	20		

**Table 2 animals-09-01082-t002:** Genotypic and allelic frequencies (%) of bovine *PLAG1* gene.

Breed	Genotype			Samples	Genotype Frequencies			Allelic Frequencies		HWE
	WW	WD	DD	*N*	*p*-_WW_	*p*-_WD_	*p*-_DD_	*p* _-W_	*p* _-D_	*p*-Value
YL	154	223	74	451	0.34	0.5	0.16	0.59	0.41	0.65

WW, indel wild type; WD, heterozygous type; DD, deleted type. *N*, cattle number. HWE, Hardy–Weinberg equilibrium χ^2^ value. YL, Yunling.

**Table 3 animals-09-01082-t003:** Genetic diversity parameters in YL.

Breed	Ne	PIC	He
YL	1.94	0.37	0.48

Ho, gene homozygosity; Ne, effective allele numbers; PIC, polymorphism information content.

**Table 4 animals-09-01082-t004:** Statistical association analysis of bovine *PLAG1* gene indel with growth traits in Yunling.

Growth Traits	Genotype (Mean ± SE)			*p*-Value
	WW	WD	DD	
Body height (cm)	129.17 ± 5.86 ^b^	130.41 ± 5.49 ^a^	128.95 ± 4.91 ^b^	0.041 *
Hip cross height (cm)	131.65 ± 15.06 ^b^	134.26 ± 5.65 ^a^	132.61 ± 5.31 ^ab^	0.039 *
Body length (cm)	155.92 ± 15.70	154.39 ± 13.78	153.96 ± 8.93	0.482
Heart girth (cm)	196.58 ± 12.65 ^ab^	197.91 ± 10.99 ^a^	193.95 ± 11.42 ^b^	0.039 *
Chest width (cm)	49.24 ± 5.48	49.27 ± 4.95	48.89 ± 5.00	0.855
Rump length (cm)	50.51 ± 3.24	50.36 ± 3.79	49.65 ± 3.66	0.224
Chest depth (cm)	68.95 ± 6.06	68.73 ± 5.94	67.85 ± 5.24	0.408
Hip width (cm)	57.80 ± 5.79	57.96 ± 5.78	56.73 ± 6.33	0.288
Hucklebone width (cm)	22.47 ± 2.30	22.29 ± 2.49	22.19 ± 1.95	0.631
Hip circumference (cm)	113.37 ± 12.25	112.66 ± 8.83	112.73 ± 8.34	0.787

LSM ± SE, least-squares means and their standard errors for each genotypic class reported. ^a,b^ Means significantly different for genotype frequencies with genetic groups (*, *p* < 0.05).

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
