# Peer review of "The Distribution Characteristics of a 19-bp Indel of the PLAG1 Gene in Chinese Cattle"

_animals, 2019, doi:10.3390/ani9121082_

Round 1

Reviewer 1 Report

Dear Authors,

This is impossible to make any conclusions without table with correlation between physical metrics and genotypes. 

"These results demonstrated that the 19bp-Indel of the PLAG1 gene is 208 an effective marker for height trait, and this marker can be used as a candidate molecular marker for 209 bovine breeding." is unproven statement. And this marker is too short for useful marker.

And you have to proofread the text carefully.

Other found problems:

75 missed space and many other places

86 please expand "so on" and provide a table with these values in the supplementary table and with genotypes

102 you use percent not frequency 

127 Figure 1 and 2 is too small, and it is impossible to review it.

To improve your work, I suggest checking these variants in genomes of several other species of Bovidae that different in body size.

Author Response

1:This is impossible to make any conclusions without table with correlation between physical metrics and genotypes.

Response 1: We are grateful for the suggestion. I have collected the body height of as many breeds of cattle as possible in Animal genetic resources in China-bovines and added these data to the Table S2.

2: "These results demonstrated that the 19bp-Indel of the PLAG1 gene is 208 an effective marker for height trait, and this marker can be used as a candidate molecular marker for 209 bovine breeding." is unproven statement. And this marker is too short for useful marker.

Response 2: This Indel (rs523753416) has been confirmed in previous studies. And we have expanded the sample size and the coverage of breeds and regions in our research. This marker can be used as a reference marker for the selection of body size traits. I have modified the absoluteness of this sentence. "These results sugguested that the 19-bp Indel of the PLAG1 gene is an effective marker for growth traits in Chinese cattle."

3:75 missed space and many other places.

Response 3: The space has been added, and we checked the missing of many other spaces.

4:86 please expand "so on" and provide a table with these values in the supplementary table and with genotypes

Response 4: We have expanded “so on”. And in Table 4, we provided the association analysis of bovine PLAG1 gene indel with growth traits and genotypes in Yunling.

Table 4. Statistical association analysis of bovine PLAG1 gene indel with growth traits in YUNLING.

Growth traits

GenotypeMean±SE

P value

WW

WD

DD

Body height (cm)

129.17±5.86b

130.41±5.49a

128.95±4.91b

0.041*

Hip cross height

(cm)

131.65±15.06b

134.26±5.65a

132.61±5.31ab

0.039*

Body length(cm)

155.92±15.70

154.39±13.78

153.96±8.93

0.482

Heart girth(cm)

196.58±12.65ab

197.91±10.99a

193.95±11.42b

0.039*

Chest width (cm)

49.24±5.48

49.27±4.95

48.89±5.00

0.855

Rump length (cm)

50.51±3.24

50.36±3.79

49.65±3.66

0.224

Chest depth (cm)

68.95±6.06

68.73±5.94

67.85±5.24

0.408

Hip width(cm)

57.80±5.79

57.96±5.78

56.73±6.33

0.288

Hucklebone width

(cm)

22.47±2.30

22.29±2.49

22.19±1.95

0.631

Hip circumference(cm)

113.37±12.25

112.66±8.83

112.73±8.34

0.787

5:102 you use percent not frequency

Response 5: We are very sorry and have corrected my mistake.

6:127 Figure 1 and 2 is too small, and it is impossible to review it.

Response 6: We are sorry for these figures, and we have adjusted the clarity of these figures.

(a)(b)

Figure 1.(a) The result of polyacrylamide gel electrophoresis. (b) The results of sanger sequencing.

Figure 2. Distribution of allele frequency of the PLAG1 gene in Chinese breeds. BH, Bohai Black; BS, Bashan; DBS, Dabeishan; DH, dehong;DQ, Diqing;DZ, Dianzhong; GF, Guangfeng; GL, Guanling; GZWN, Weining; JA, Ji’an; JC, Jiangcheng; JJ, Jinjiang; JN, Jinnan; JX, Jiaxian red; KZ, Kazakh; LX, Luxi; MG, Mongolian; ND, Nandan; NY, Nanyang; QC, Qinchuan; SJ, Sanjiang; SH, Shigatse Humped; SJ, Sanjiang; SN, sinan; TB, Tibetan; WC, Wuchuan; WL, Wuling; WN, Wannan; WS, Wenshan; XN, xianan;YB, Yanbian; YL, Yunling; ZB, Zaobei; ZS, Zaosheng; ZT, Zhaotong

7:To improve your work, I suggest checking these variants in genomes of several other species of Bovidae that different in body size.

Response 7: We are grateful for the suggestion. In this study, we selected a total of 38 cattle breeds, which basically covered cattle breeds in various regions of China. These breeds have different body size traits, which we consider to be representative.

Reviewer 2 Report

The aim of the research described in this paper is really interesting and the methods used are appropriate. Results are useful and Discussion and Conclusions are congruent. For this reason I recommend the work for the publication. However the whole article needs to be widely revised. In fact:

1) the meaning of many sentences is missing or incorrect (for ex.: LINE 13-14; LINE 14-15; LINE 22; …….)

2) Some statements need a Reference (for ex.: LINE 50à52 ….. )

3) Some references do not support what is stated in the sentences with which they are associated (for ex.: LINE 55; LINE 63; LINE 68….)

4) LINE 80 --> What does “CULTIVATES” Stand for? I’m unable to understand what breeds they belong.

5) In the section Meterials and Methods it is necessary to indicate brand and type of all the products employed in the work specifying factory and catalog number.

6) In the section Meterials and Methods it is necessary to better specify DNA extraction methods, and all the methods used for the electrophoresis on the polyacrilammide gel (composition, type of chamber, staining tecnique….)

7) LINE 89: specify the accession number of the sequence used to generate the primers and the software used.

8) Please indicate the machine used for PCR (Factory) and the sequencer machine used by Shanghai Sangon Biotech for sequencing.

9) The figures are too small so I’m unable to analyse them.

10) References must be formatted according to the journal’s instructions.

11) The cited bibliography should be extended to better describe the gene and its functions and the importance that it could have from a zootechnical point of view and in Cattle breeding.

Throughout the paper there are some mistakes like:

LINE 14: Are them 38 or 37 the cattle breeds that have been analysed?

LINE 26: Please check the Accession Number, it refers to the whole chromosome, but I found that there is the accession number for the PLAG1 gene in Bos Taurus, didn’t you use this one for your work?

LINE 54: there is an extraspace after comma.

LINE 78: Are them Cattle breeds or Cattle Varieties? (The terms are not equivalent)

Table 4 The heading “Type” should be substituted with “Genotype”

Author Response

[Comment of Reviewer 2:]

1) the meaning of many sentences is missing or incorrect (for ex.: LINE 13-14; LINE 14-15; LINE 22; …….)

Response 1: Thank you for your precious advice.We have corrected these sentences.

2) Some statements need a Reference (for ex.: LINE 50à52 ….. )

Some references do not support what is stated in the sentences with which they are associated (for ex.: LINE 55; LINE 63; LINE 68….)

Response 2,3: We are sorry that there are many questions in the references in the previously manuscript version. We have now revised the references, including the addition of some references and changes in location.

LINE 80 --> What does “CULTIVATES” Stand for? I’m unable to understand what breeds they belong.

Response 4: We have changed “cultivates breeds” to “commercial breeds”.

5) In the section Meterials and Methods it is necessary to indicate brand and type of all the products employed in the work specifying factory and catalog number.

Response 5: We have investigated and indicated the source factory, brand and type of all the products.

6 In the section Meterials and Methods it is necessary to better specify DNA extraction methods, and all the methods used for the electrophoresis on the polyacrilammide gel (composition, type of chamber, staining tecnique….)

Response 6: We have chosen to add two references, which introduce DNA extraction and polyacrylamide gel electrophoresis. We also added a brief description in the article.

7) LINE 89: specify the accession number of the sequence used to generate the primers and the software used.

Response 7: We have added the accession number of the sequence used to generate the primers and the software used.“The insertion/deletion site (No. AC_000171.1 25020872-25020890) was genotyped using a pair of primers. The primers were designed on the NCBI Primer-BLAST and synthesized by Shanghai Sangon Biotech Engineering Co., Ltd. The purification method is HAP. Provide a total of 2OD and divided into two tubes.”

8) Please indicate the machine used for PCR (Factory) and the sequencer machine used by Shanghai Sangon Biotech for sequencing.

Response 8: The instrument for sequencing is ABI 3730xl in Shanghai Sangon Biotech. We have indicated the machine in our manuscript.

9) The figures are too small so I’m unable to analyse them.

We are sorry for these figures,and we have adjusted the clarity of these figures.

(a)(b)

Figure 1.(a) The result of polyacrylamide gel electrophoresis. (b) The results of sanger sequencing.

Figure 2. Distribution of allele frequency of the PLAG1 gene in Chinese breeds. BH, Bohai Black; BS, Bashan; DBS, Dabeishan; DH, dehong;DQ, Diqing;DZ, Dianzhong; GF, Guangfeng; GL, Guanling; GZWN, Weining; JA, Ji’an; JC, Jiangcheng; JJ, Jinjiang; JN, Jinnan; JX, Jiaxian red; KZ, Kazakh; LX, Luxi; MG, Mongolian; ND, Nandan; NY, Nanyang; QC, Qinchuan; SJ, Sanjiang; SH, Shigatse Humped; SJ, Sanjiang; SN, sinan; TB, Tibetan; WC, Wuchuan; WL, Wuling; WN, Wannan; WS, Wenshan; XN,xianan;YB, Yanbian; YL,Yunling; ZB, Zaobei; ZS, Zaosheng; ZT, Zhaotong

10) References must be formatted according to the journal’s instructions.

Response 10: Sorry for the format. We have made changes to the references.

11) The cited bibliography should be extended to better describe the gene and its functions and the importance that it could have from a zootechnical point of view and in Cattle breeding.

Response 11: Thank you for your advice. We are sorry that there are many questions in the references in the previously manuscript version. We have now revised the references, including the addition of some references and changes in location.

Throughout the paper there are some mistakes like:

LINE 14: Are them 38 or 37 the cattle breeds that have been analysed?

Sorry for typing. We analysed 37 cattle breeds. We have changed the 38 to 37.

LINE 26: Please check the Accession Number, it refers to the whole chromosome, but I found that there is the accession number for the PLAG1 gene in Bos Taurus, didn’t you use this one for your work?

The Accession Number we used is for the PLAG1 gene in Bos Taurus. This is a screenshot of the Accession Number we used in the website. (https://www.ncbi.nlm.nih.gov/gene/539210)

LINE 54: there is an extra space after comma.

We have deleted the extra space.

LINE 78: Are them Cattle breeds or Cattle Varieties? (The terms are not equivalent)

We have changed all the “varieties” to “breeds”.

Table 4 The heading “Type” should be substituted with “Genotype”

We have changed the “Type” to “Genotype”.

Round 2

Reviewer 1 Report

Thank for the manuscript improving.

Author Response

Thank you for helping us review our manuscript.

Reviewer 2 Report

Dear Authors,

The changes made did not resolve all the critical issues previously indicated. I suggest you review the article in order to improve the English form and to pay attention to the statements made and to the references used in support. Here are some necessary corrections. 

Major items:

1) In “Simple summary” the definition of InDel is not correct, moreover the text is repetitive, please reformulate better all the paragraph.

2) References 4,6,7,10, 18, 20, 21, 22, 24, do not fit the sententence.

3) Reference 11 and 14 are not necessary.

4) Reference 15: I’m unable to verify the reference since it is difficult to trace the article in the absence of the name of the journal and the volume.

5) Figures are still very small so I’m unable to analyse them

6) The References are still not formatted as required by journal rules.

Minor items and suggestions

1) LINE 53-54: According to reference 3 the PLAG1 gene does not encodes “a ubiquitinated and phosphorylated transcription factor IGF2” but “oncogenic capability of PLAG1 is mediated, at least partly, by the IGF-II mitogenic signaling pathway” and “IGF-II expression can be highly stimulated by PLAG1. Moreover, a drastic up-regulation of IGF-II promoter 3 transcripts coincides with PLAG1 activation in human pleomorphic adenomas of the salivary glands, indicating a correlation between PLAG1 and IGF-II expression”.

Please rewrite the sentence.

2) LINE 91-92: “The primer sequence, primer size,…” please change with “Size and sequence of the primers,….

3) LINE 24-25: “There was a 19-bp insertion/deletion (Indel) in the first intron of the bovine PLAG1 gene (GenBank Accession No. AC_000171.1).”

Please change with:

In Cattle a 19-bp insertion/deletion (Indel) has been identified in the intron 1 of PLAG1 gene (GenBank Accession No. AC_000171.1).

4) LINE 28-29: “There are differences in height among cattle breeds of different geographical distribution in China.”

Please change with:

Chinese Cattle breeds show a difference in height related to geographical distribution

5) LINE 65: please replace “widely found” with “widespread”.

6) LINE 68-70: Please check the sentence and improve its English form.

7) LINE 128-129: “A total of ten PCR products were selected and sent to Shanghai Sangon Biotech for sequencing. The sequencing results are shown in Fig. 1b.” Please eliminate both sentences, they are a repetitions of methods.

8) LINE 141: Please replace the word “cultivar” with the word “breed”

9) LINE 154-156: Please check the sentence, the meaning is not clear to me.

10) LINE 170: Please check the sentence, the meaning is not clear to me.

11) LINE 173: Please replace “identified” with “analysed”

12) LINE 174: Please replace “introns contain” with “they contain”

13) Change “Intron mutations” with “intronic mutations” throughout the text.

14) LINE 197-198:” the polymorphism of this polymorphism…” replace with “the variants of this polymorphism”

15) LINE 210-211: “we found that in Yunling cattle that the 19 bp-indel polymorphism of PLAG1…” replace with “we found that, in Yunling cattle, the 19 bp-indel polymorphism of PLAG1…”

Author Response

[Comment of Reviewer 2:]

Major items:

1) In “Simple summary” the definition of InDel is not correct, moreover the text is repetitive, please reformulate better all the paragraph.

Thank you for the advice. We have rewrited all the paragraph.

2) References 4,6,7,10, 18, 20, 21, 22, 24, do not fit the sententence.

Sorry for our mistakes. These references have been changed.

3) Reference 11 and 14 are not necessary.

We have deleted reference 11 and 14.

4) Reference 15: I’m unable to verify the reference since it is difficult to trace the article in the absence of the name of the journal and the volume.

We have changed the reference with a better one to introduce the method of polyacrylamide gel electrophoresis.

5) Figures are still very small so I’m unable to analyse them

Sorry for the small pictures again. We have enlarged the figures.

6) The References are still not formatted as required by journal rules.

We used EndNote software to prepare our references as required by MDPI rules. Please help us to confirm the format.

Minor items and suggestions

1) LINE 53-54: According to reference 3 the PLAG1 gene does not encodes “a ubiquitinated and phosphorylated transcription factor IGF2” but “oncogenic capability of PLAG1 is mediated, at least partly, by the IGF-II mitogenic signaling pathway” and “IGF-II expression can be highly stimulated by PLAG1. Moreover, a drastic up-regulation of IGF-II promoter 3 transcripts coincides with PLAG1 activation in human pleomorphic adenomas of the salivary glands, indicating a correlation between PLAG1 and IGF-II expression”.

Please rewrite the sentence.

Sorry for our mistakes. We have rewrited the sentence. “Pleomorphic adenoma gene 1 (PLAG1) is involved in cell proliferation by directly regulating a wide array of target genes, including a number of growth factors such as IGF2. “

2) LINE 91-92: “The primer sequence, primer size,…” please change with “Size and sequence of the primers,….

Thank you. The sentence has been changed.

3) LINE 24-25: “There was a 19-bp insertion/deletion (Indel) in the first intron of the bovine PLAG1 gene (GenBank Accession No. AC_000171.1).”

Please change with:

In Cattle a 19-bp insertion/deletion (Indel) has been identified in the intron 1 of PLAG1 gene (GenBank Accession No. AC_000171.1).

We have changed the sentence.

4) LINE 28-29: “There are differences in height among cattle breeds of different geographical distribution in China.”

Please change with:

Chinese Cattle breeds show a difference in height related to geographical distribution

We have changed the sentence.

5) LINE 65: please replace “widely found” with “widespread”.

The words have been changed.

6) LINE 68-70: Please check the sentence and improve its English form.

We have changed the sentence: “Based on this background, whether the 19-bp Indel affected in body height between northern and southern cattle breeds in China was the aim in our study.”

7) LINE 128-129: “A total of ten PCR products were selected and sent to Shanghai Sangon Biotech for sequencing. The sequencing results are shown in Fig. 1b.” Please eliminate both sentences, they are a repetitions of methods.

We have deleted the sentences.

8) LINE 141: Please replace the word “cultivar” with the word “breed”

The word “cultivar” has been replaced with “breed”.

9) LINE 154-156: Please check the sentence, the meaning is not clear to me.

Sorry for the sentence. We have changed the sentence : ”According to the Botstein polymorphic information content classification standard, Yunling cattle had medium genetic diversity in the indel locus.”

10) LINE 170: Please check the sentence, the meaning is not clear to me.

Sorry for the sentence. We have changed the sentence : ”Note: LSM±SE, least squares mean and their standard errors for each genotypic class reported. a,b Means significantly different for genotype frequencies with genetic groups (P < 0.05).”

11) LINE 173: Please replace “identified” with “analysed”

The word “identified” has been replaced with “analysed”.

12) LINE 174: Please replace “introns contain” with “they contain”

The words “introns contain” has been replaced with “they contain”.

13) Change “Intron mutations” with “intronic mutations” throughout the text.

We have changed “Intron mutations” with “intronic mutations” throughout the text.

14) LINE 197-198:” the polymorphism of this polymorphism…” replace with “the variants of this polymorphism”

We have changed the words.

15) LINE 210-211: “we found that in Yunling cattle that the 19 bp-indel polymorphism of PLAG1…” replace with “we found that, in Yunling cattle, the 19 bp-indel polymorphism of PLAG1…”

Thank you. The sentence has been changed.
